# Induction of human somatostatin and parvalbumin neurons by expressing a single transcription factor LIM homeobox 6

Fang Yuan[1,2], Xin Chen[3], Kai-Heng Fang[1], Yuanyuan Wang[4], Mingyan Lin[4], Shi-Bo Xu[1], Hai-Qin Huo[1], Min Xu[1], Lixiang Ma[5], Yuejun Chen[6], Shuijin He[3]*, Yan Liu[1,2,7]*

[1]Institute for Stem Cell and Neural Regeneration, School of Pharmacy, Nanjing Medical University, Nanjing, China; [2]State Key Laboratory of Reproductive Medicine, Nanjing Medical Unveristy, Nanjing, China; [3]School of Life Science and Technology, ShanghaiTech University, Shanghai, China; [4]Department of Neuroscience, School of Basic Medical Sciences, Nanjing Medical University, Nanjing, China; [5]Department of Human Anatomy and Histology, Institute of Stem Cells and Regenerative Medicine, Fudan University Shanghai Medical School, Shanghai, China; [6]Institute of Neuroscience, Chinese Academy of Sciences, Beijing, China; [7]Institute for Stem Cell and Regeneration, Chinese Academy of Sciences, Beijing, China

*For correspondence:
yanliu@njmu.edu.cn (YL);
heshj@shanghaitech.edu.cn (SH)

**Competing interests:** The authors declare that no competing interests exist.

**Abstract** Human GABAergic interneurons (GIN) are implicated in normal brain function and in numerous mental disorders. However, the generation of functional human GIN subtypes from human pluripotent stem cells (hPSCs) has not been established. By expressing LHX6, a transcriptional factor that is critical for GIN development, we induced hPSCs to form GINs, including somatostatin (SST, 29%) and parvalbumin (PV, 21%) neurons. Our RNAseq results also confirmed the alteration of GIN identity with the overexpression of *LHX6*. Five months after transplantation into the mouse brain, the human GABA precursors generated increased population of SST and PV neurons by overexpressing *LHX6*. Importantly, the grafted human GINs exhibited functional electrophysiological properties and even fast-spiking-like action potentials. Thus, expression of the single transcription factor LHX6 under our GIN differentiation condition is sufficient to robustly induce human PV and SST subtypes.
DOI: https://doi.org/10.7554/eLife.37382.001

## Introduction

GABAergic interneurons (GINs) form the major inhibitory system in the mammalian cortex (*Molyneaux et al., 2007*; *Wonders and Anderson, 2006*). There are many subtypes of cortical GINs that are differentiated on the basis of the expression of molecular markers, including calbindin (CB), calretinin (CR), parvalbumin (PV), and somatostatin (SST) (*Wonders and Anderson, 2006*). PV interneurons display a chandelier or basket morphology and have a fast-spiking property. They are essential for regulating emotion and learning (*Hu et al., 2014*). SST interneurons are characterized as Martinotti or non-Martinotti in morphology and play important roles in reducing epileptic form activity and in facilitating motor and spatial learning (*Yavorska and Wehr, 2016*). Dysfunction of PV and SST interneurons is involved in neurological and psychiatric diseases such as epilepsy, schizophrenia

and autism (*Fee et al., 2017*; *Marín, 2012*; *Hashimoto et al., 2003*). Thus, the availability of human PV and SST GINs is important for cell-replacement treatment and for studies of disease etiology.

Cortical GINs are primarily derived from the MGE (medial ganglionic eminence), which is located in the ventral forebrain during embryonic development (*Kelsom and Lu, 2013*; *Hansen et al., 2013*; *Ma et al., 2013*). The developmental process is orchestrated by a cascade of transcription factors, including DLX2, DLX5, NKX2.1, and LHX6, which is induced by the morphogen Sonic hedgehog (SHH) that is secreted by the notochord (*Kelsom and Lu, 2013*; *Xu et al., 2005*; *Xu et al., 2008*; *Liodis et al., 2007*; *Flandin et al., 2011*; *Du et al., 2008*). On the basis of this principle, approaches have been developed for generating human GINs from hPSCs by efficiently inducing MGE progenitors using sonic hedgehog (SHH) (*Liu et al., 2013a*), or SHH together with telencephalic inducers (*Maroof et al., 2013*; *Nicholas et al., 2013*; *Kim et al., 2014*). However, GINs expressing SST and PV are usually generated only after 100 days and relied on co-culture with rodent cortical neurons/ astrocytes and cell sorting (*Maroof et al., 2013*; *Nicholas et al., 2013*). Recently, GINs have also been generated by overexpressing transcription factors, such as Ascl1 and Dlx2. Again, however, the populations of SST(~9%) and PV neurons are very low after 5 weeks of culture (*Yang et al., 2017*). Thus, efficient generation of human PV and SST interneurons from hPSCs is desirable for disease modeling using patient induced pluripotent stem cells ( iPSCs).

In this study, we established human embryonic stem cell (ESC) and iPSC lines with inducible expression of *LHX6*. Differentiation of transgenic *LHX6* hPSCs to GINs under our established GIN differentiation protocol (*Yuan et al., 2015*) significantly increased the percentage of PV and SST interneuron subtypes within 80 days when LHX6 is induced. Importantly, the PV and SST neurons that were generated following transplantation into the mouse brain exhibit increased population size and a fast-spiking-like electrophysiological property.

## Results

### Establishment of inducible *LHX6* overexpressing hPSC cell lines

We first established a human ESC (H9) line and an iPSC line (ihtc) with inducible expression of *LHX6* by inserting, using TALEN-mediated targeting, a tet-on inducible cassette into the AAVS1 site (*Qian et al., 2014*). After electroporation, transgenic LHX6 hPSCs were selected by puromycin (*Figure 1a*). The transgenic colonies showed a morphology similar to that of the parental PSCs (*Figure 1b*). For the H9 cell line, 14 colonies were selected by puromycin treatment. And quantative real-time PCR (qPCR) experiments were performed to detect the expression levels of *LHX6* mRNA after 3 days continuous induction with doxycycline (dox), which turns on the expression of LHX6 from the promoter. After induction, three of the 14 colonies (efficiency ~21%) showed high expression of *LHX6* when compared with non-induced colonies. Furthermore, the expression of *LHX6* was confirmed in one of these colonies (H9-01) by LHX6 immunostaining. The same experiment was performed on the ihtc cell line, and the ihtc-03 colony and two of eight colonies were shown to overexpress *LHX6* (efficiency ~25%). The overexpression of *LHX6* mRNA in colony ihtc-03 was confirmed by assessing the expression of LHX6 protein (*Figure 1c*). The H9-01 and ihtc-03 cell lines were then cultured and expanded for further experiments.

### Overexpression of LHX6 biases dorsal forebrain precursors to the ventral fate

In the absence of exogenous morphogens, human PSCs differentiate to a nearly uniform population of neural precursors with the dorsal forebrain identity (*Li et al., 2009*). We asked whether expression of LHX6 alters the identity of differentiated progenitors. When the transgenic hPSCs were differentiated to neural progenitors under the 'default' condition for 17 days (*Figure 1d*), the mRNA levels of the ventral transcription factors *LHX6*, *LHX8*, *NKX2.1*, and *SHH* were significantly increased, whereas the level of the dorsal transcription factor PAX6 decreased in the neural progenitors when *LHX6* was induced (*Figure 1e*).

Immunostaining of the neural precursors at day 25 indicated that both the LHX6-expressing and the parental PSC-derived neural precursors were positive for FOXG1 (*Figure 1f*), indicating that the expression of LHX6 does not alter the forebrain identity. Among *LHX6* OE (*LHX6* overexpression cells), however, the population of dorsal precursors that express PAX6 was decreased (H9-01 —

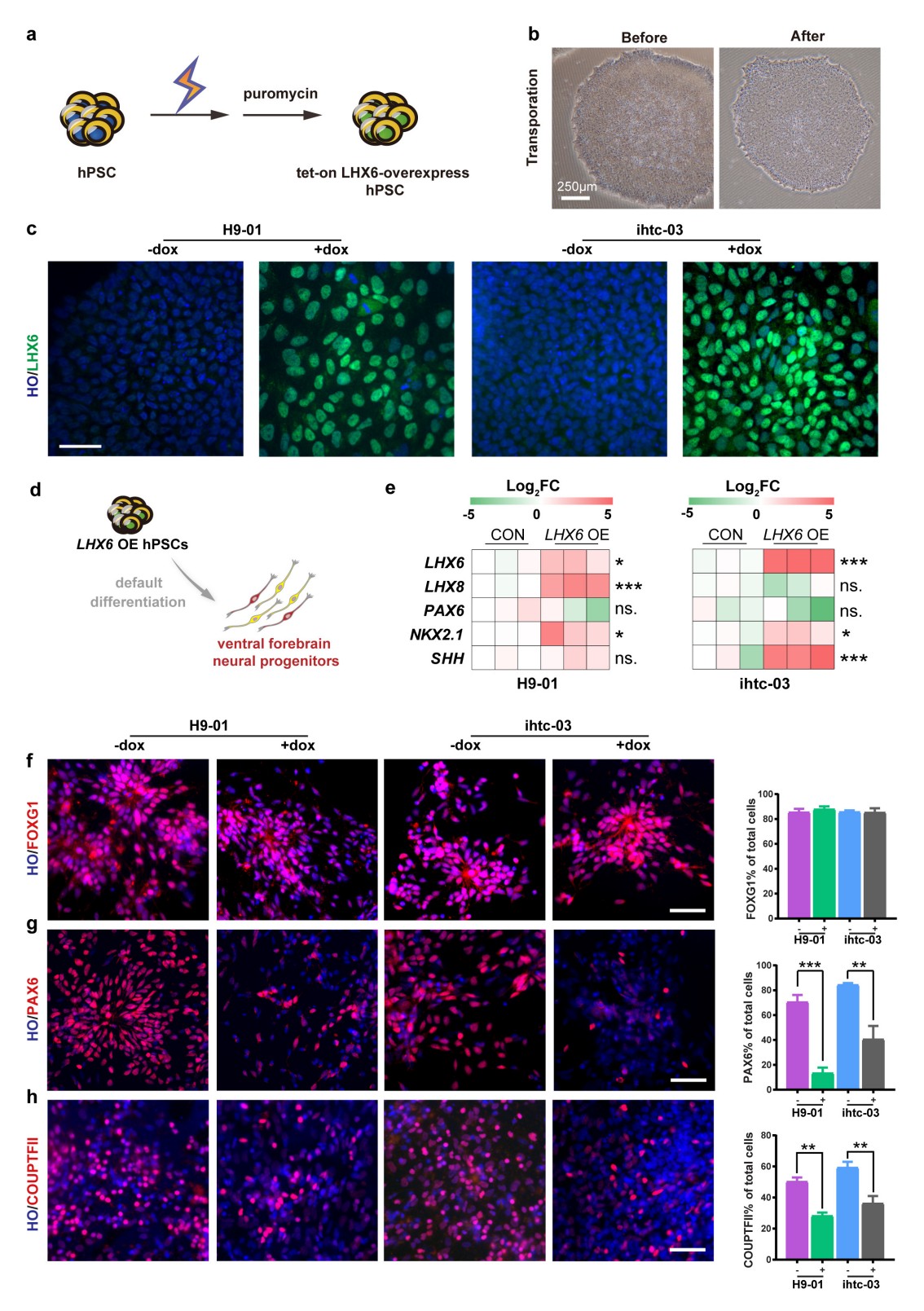

**Figure 1.** Construction of inducible *LHX6* OE hPSCs. (a) Schematic representation of electroporation to establish inducible *LHX6* overexpressing (OE) hPSCs. (b) Bright-field images of hPSC colonies before and after electroporation. (c) After doxycycline induction, two inducible *LHX6* OE hPSC cell lines expressed LHX6. Scale bar, 50 μm. (d) Schematic showing the differentiation of transgenic hPSC lines into dorsal neurons without adding morphogens. CON: default control group (−dox), *LHX6* OE: *LHX6* OE group (+dox). (e) mRNA expression levels for two transgenic hPSC-derived neurospheres and

*Figure 1 continued on next page*

*Figure 1 continued*

each control at day 17; n ≥ 3 for each cell line. (**f–h**) Representative images and quantification of transcription factors FOXG1 (**f**), PAX6 (**g**) and COUPTFII (**h**) expressed in CON and *LHX6* OE neural precursors from two cell lines.

DOI: https://doi.org/10.7554/eLife.37382.002

13% +dox vs 71% –dox; ihtc-03 — 37% +dox vs 84% –dox) (*Figure 1g*). The population of cells that express the caudal ganglionic eminence marker COUPTFII was dramatically decreased by *LHX6* overexpression (28% of the H9-01 *LHX6* OE vs. 50% of controls, and 36% of the ihtc-03 *LHX6* OE vs. 59% of controls) (*Figure 1h*). Interestingly, Nkx2.1, a principle transcription factor that is involved in the specification of MGE progenitors (*Xu et al., 2008*; *Du et al., 2008*), was not detected in *LHX6* OE groups. This may be explained by the fact that studies in mice have shown that *Lhx6* lies downstream of *Nkx2.1* (*Elias et al., 2008*). Together, the results indicate that *LHX6* overexpression biases the forebrain neural progenitors to the ventral identity under the default differentiation condition.

## *LHX6* promotes the generation of GINs

As differentiation to neurons progresses, or at day 35 from hPSC differentiation, the percentage of GABA-positive neurons in the *LHX6* OE group was twice that in the control group (22% vs. 11% in H9-01 cells and 16% vs. 7% in ihtc-03 cells) (*Figure 2a*). Under the default differentiation scheme (without the presence of exogenous ventral inducers), there are usually no SST- or PV-positive neurons (*Li et al., 2009*). Indeed, we observed a few CB and CR neurons but no SST or PV neurons in the control group. By contrast, there are sizeable populations of SST and PV neurons, as well as CR, CB, and nNOS cells, in the *LHX6* group (*Figure 2b and c*). Importantly, SST and PV neurons exhibited a more complex morphology, resembling multipolar cells (*Figure 2d*). We set the criterion of neurons having more than five secondary branches as morphology characteristic of SST and PV neurons. Most SST neurons have the characteristic morphology from day 45. About 30% of PV neurons have this characteristic morphology at day 80. Therefore, *LHX6* is sufficient to promote the differentiation of GINs, including SST and PV subtypes, without the presence of SHH.

## *LHX6* enhances the generation of SST and PV neurons

Highly enriched GINs can be generated from hPSCs by patterning the progenitors with SHH, although the generation of subtypes, especially of SST and PV neurons, takes a long time and the efficiency is low (*Liu et al., 2013a*; *Maroof et al., 2013*; *Cunningham et al., 2014*). Given that *LHX6* expression alone is sufficient to generate SST and PV neurons within 5 weeks of default differentiation, we asked whether *LHX6* further enhances the generation of human SST and PV neurons when the progenitors are ventralized. We differentiated hPSCs into MGE-like progenitors in the presence of SAG, a small molecule that activates SHH signaling (*Figure 3a*). Upon induction of *LHX6* expression with dox (ventral *LHX6* overexpression group: V-*LHX6* OE; ventral patterning group without dox: V-CON), the mRNA levels of *LHX8* and *SHH*, but not *NKX2.1*, were significantly increased in the two cell lines at day 17 (*Figure 3b*). Later, at day 25, we further confirmed *LHX6* overexpression during the differentiation process and found that *LHX6* OE groups expressed LHX6, while there was no expression of LHX6 in control groups (*Figure 3c*), as *LHX6* is a post-mitotic marker that begins to be expressed at a relatively mature neuron stage (*Vogt et al., 2014*). With the activation of SHH signaling by SAG treatment, more than 80% of the cells in both the control and *LHX6* OE groups showed expression of the MGE marker NKX2.1 at similar levels (*Figure 3c and d*). Similarly, the controls and *LHX6* OE cell lines showed no significant difference in the expression of FOXG1 or OLIG2 (*Figure 3—figure supplement 1a and b*). Also, TUJ-1 and GABA expression did not differ between the V-CON and the V-*LHX6* OE group. All groups generated similar proportions of GINs (about 80%) at day 35 (*Figure 3c and e*). These results indicate that *LHX6* OE does not alter the general GIN neurogenesis at an early stage.

When examining the GIN subtypes, we found that at day 35, CB was expressed in 18% of total cells compared with 22% of cells in the control, with no significant difference between the two (*Figure 3f*). The CR- expressing population was maintained at 20–30% from day 35 to day 80 (*Figure 3f*). SST+ were initiated at day 45 with the *LHX6* overexpression and PV+ neurons began to express at day 60. Notably, SST+ neurons dramatically increased to 29% of all neurons in the *LHX6*

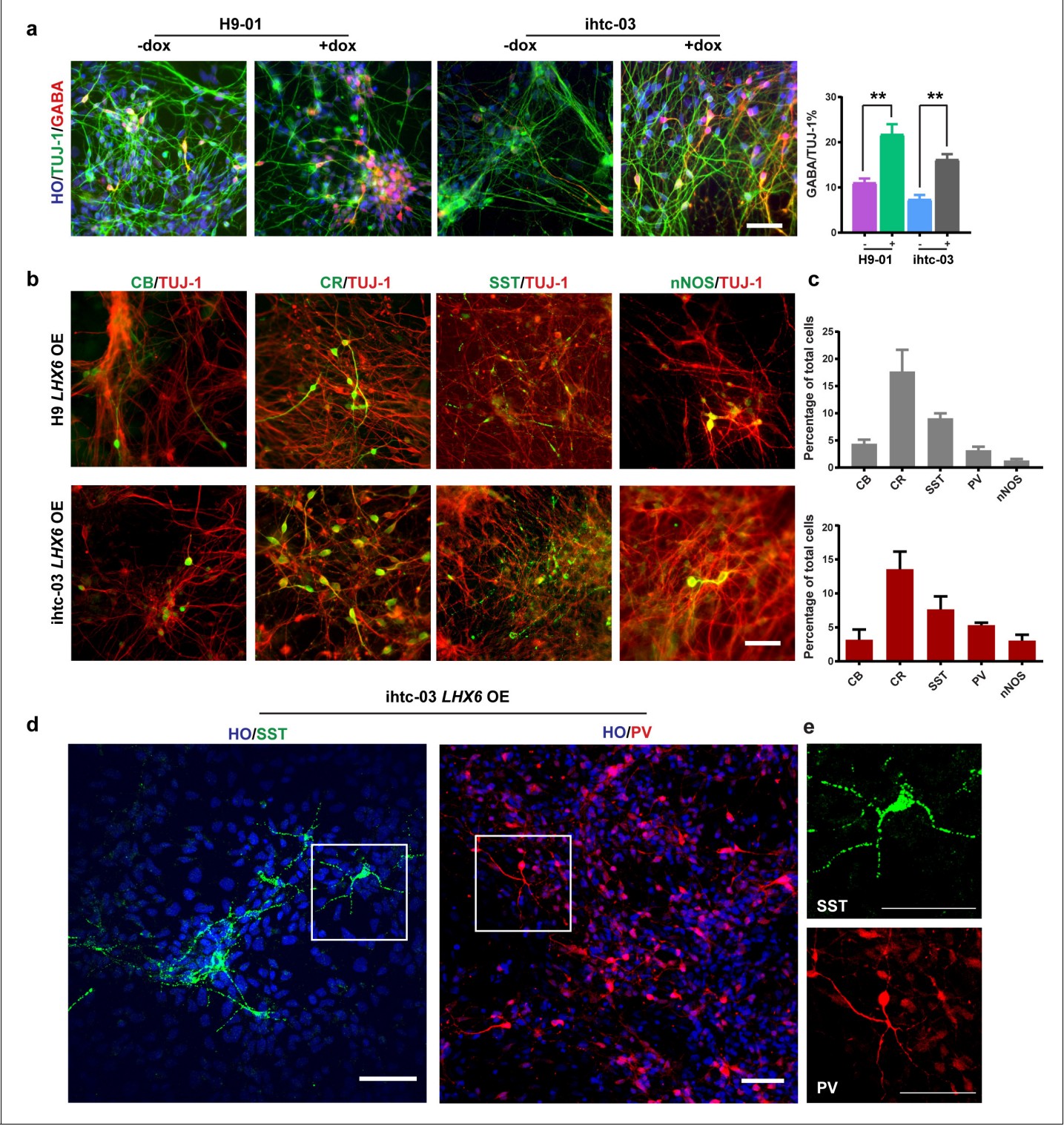

**Figure 2.** *LHX6* is sufficient to convert hPSC-derived dorsal neurons to GIN subtypes. (**a**) At day 35, immunostaining of TUJ-1 showed a similar efficiency of neuronal differentiation in all *LHX6* OE groups and controls and a higher percentage of GABA+ cells in *LHX6* OE cells. Scale bar, 50 μm. (**b**) The GIN subtypes calbindin (CB, day 35), calretinin (CR, day 45), somatostatin (SST, day 50), nNOS (day 80) were presented in the *LHX6*OE cells from two cell lines. Scale bar, 50 μm. (**c**) Quantification of CB+, CR+, SST+, and PV+ cells among the TUJ-1+ cells. Upper, H9-01 *LHX6* OE; below, ihtc-03 *LHX6* OE group. (**d**) SST and PV expression in the ihtc-03 *LHX6* OE group. (**e**) Neurons expressing SST and PV interneurons showed a characteristic morphological structure with more than five secondary branches.

DOI: https://doi.org/10.7554/eLife.37382.003

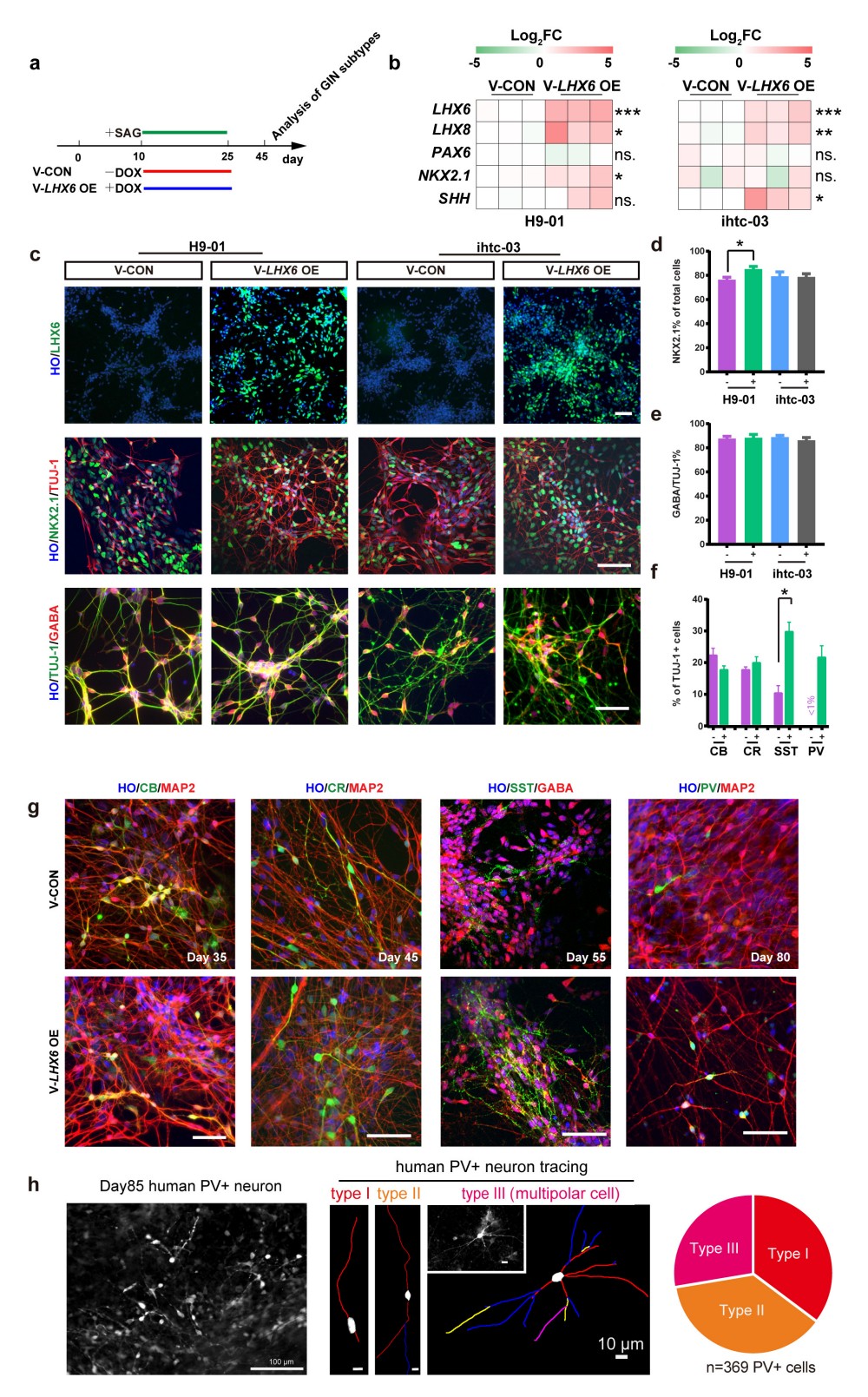

**Figure 3.** Generation of SST and PV subtypes by overexpressing *LHX6* and ventral patterning. (**a**) Schematic showing the differentiation of ventral cells from transgenic hPSC lines after treatment with SAG. V-CON, ventral control group (−dox); V-*LHX6* OE, ventral *LHX6* OE group (+dox). (**b**) The mRNA expression levels of ventral transcriptional markers in three transgenic hPSC lines and in each control at 17 days post-differentiation; n ≥ 3 biological replicates. (**c**) The proportions of LHX6+ cells in the control and *LHX6* OE groups. The NKX2.1, GABA, TUJ-1 groups all displayed high expression

*Figure 3 continued on next page*

Figure 3 continued

of LHX6 in both the H9-01 and ihtc-03 cell lines with or without dox. Scale bar, 50 μm. (d–e) Quantification of NKX2.1+ cells (d) and GINs (e). (f) Percentage of GINs in the ventral ihtc-03 *LHX6* OE group. At least 1500 cells were counted from random selected fields in each cell line, n ≥ 3 for each cell line. (g) Immunostaining of GIN subtypes in the ventral ihtc-03 control (V-CON) and *LHX6* OE groups. (h) Representative tracing images of three different types PV+ neurons (type I, 130/369; type II, 137/369; type III multipolar cells, 102/369).
DOI: https://doi.org/10.7554/eLife.37382.004
The following figure supplement is available for figure 3:

**Figure supplement 1.** GIN progenitors identity and subtype distribution.
DOI: https://doi.org/10.7554/eLife.37382.005

OE group in comparison of 12% in the control, and PV+ neurons increased to 21% in the *LHX6* OE group (*Figure 3f*) compared to <1% in the controls at day 85 (*Figure 3g and f*). Notably, the human PV+ neurons, which are induced by LHX6, pursued a multipolar structure with multiple neuritis beginning from the cell body (102 of 369 PV+ neurons) (*Figure 3h*). Similar percentages of GINs subtypes were observed in another *LHX6* OE cell line (*Figure 3—figure supplement 1c and d*).

Taken together, our results showed that *LHX6* overexpression in MGE-like progenitors did not change the ventral telencephalic fate but enhanced the inhibitory interneuron cell fate to SST+ and PV+ neurons.

## RNA-seq assay of *LHX6* OE neuron profiling

To explore the effect of *LHX6* on neural development and differentiation systematically, we performed RNA-sequencing (RNA-seq) of hPSC-derived GINs at day 50 in vitro. A total of 1303 upregulated genes and 2344 downregulated genes were determined with cutoffs of absolute $\log_2$ fold change ≥1 and adjusted p-value < 0.05 (*Figure 4a*). Among them, *LHX6* expression had 7.338 $\log_2$ fold change leading the differential genes. The genes that were upregulated were enriched in those involved in biological processes that are critical for neuron maturation and differentiation, including synaptic signaling, neurotransmission and GABAergic neuron differentiation, whereas down-regulated genes were enriched in those with functions in biological processes involved in the maintenance of conditions for early neural development, such as extracellular structure organization and angiogenesis (*Figure 4b and c*).

By comparing our expression data with those for regions of the human brain available in the Brain span Atlas of the Developing Human Brain, we confirmed that the gene expression of V-CON and V-*LHX6* OE hiPSC-derived 50-day-old neurons most closely resembles that of the human first trimester brain cortex (*Figure 4—figure supplement 1a and b*). In addition, the gene expression of the V-*LHX6* OE neurons appeared to be closer to that of to PCW 9–17 than that of the control neurons (PCW 8–9) (*Figure 4—figure supplement 1a and b*). Characterization of neural fate confirmed the inhibitory nature of our samples (*Figure 4d*), which showed the strongest correlation to neuronal subtypes of MGE origin. In comparing our data with those on neuronal transcription factors that had been identified before (*Lake et al., 2016*), we found that our samples resembled most inhibitory identities and that the *LHX6* OE neurons were closer to inhibitory plots than control group (*Figure 4d*). To characterize the neuronal fate, we compared the gene expression profiles of our samples with the interneuron classifications. At day 50, SST mRNA showed a significant increase along with LHX6 gene expression. However, PV mRNA did not show anobvious change at this time point, which is understandable given that PV generation begins at a relatively late stage when compared with SST neuron generation. Considering the immature state of our samples, these results might imply that our neurons were in a stage of early development/differentiation characterized by a high capacity for neuronal migration. Prediction of the neural functional state of our samples implied that V-*LHX6* OE neurons were more mature and functionally active than control neurons (*Bardy et al., 2016*), although neither acquires fully functional maturity in 50 days (*Figure 4e and f*).

Taken together, our RNAseq results showed that the V-*LHX6* OE hPSC-derived neurons resembled PCW 9–17 inhibitory neuronal identity, and that they were in a more mature state than the controls.

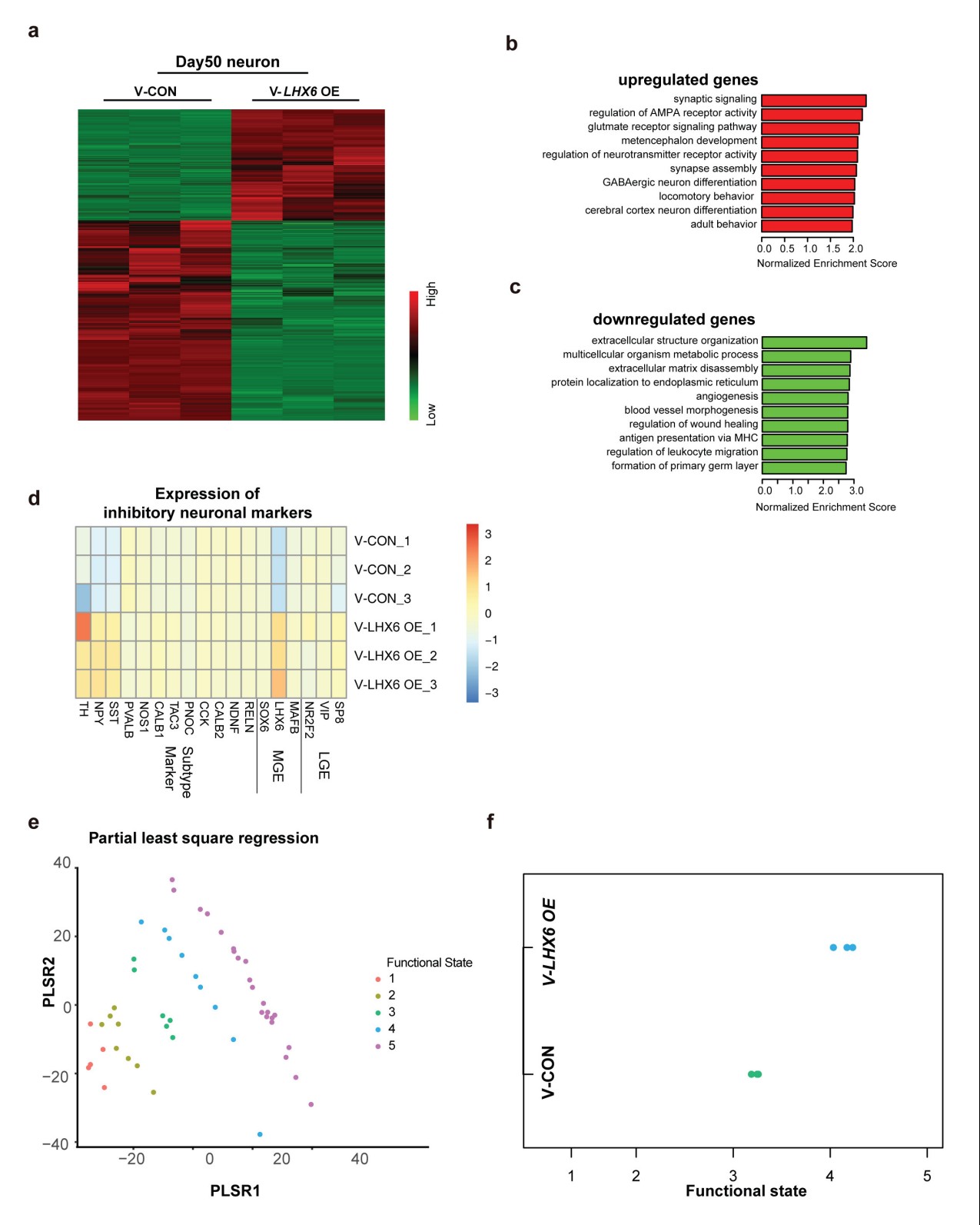

**Figure 4.** Integrative transcriptomic analyses of hPSC-derived GINs at day 50. (**a**) Heatmap showing relative expression of 3467 differentially expressed genes in V-*LHX6* OE cells compared to controls. (**b**) Bar plot presenting the top 10 non-overlapping enriched gene ontology (GO) terms in upregulated genes. (**c**) Bar plot presenting the top 10 non-overlapping enriched GO terms in downregulated genes. (**d**) Heapmap of genes associated with lateral ganglionic eminence (LGE), MGE or subtype interneuron genes in our cells (*Lake et al., 2016*). (**e**) Functional order (1–3, immature; 4, transitional and 5,

*Figure 4 continued on next page*

*Figure 4 continued*

highly functional neurons) displayed by Patch-Seq single cell population samples when projected on the first and second partial least-square (PLS) regression components of single cell RNA-seq profiles. (f) Functional state of our samples predicted by the PLSC1 and 2 derived linear model.

DOI: https://doi.org/10.7554/eLife.37382.006

The following figure supplement is available for figure 4:

**Figure supplement 1.** Comparison of hPSC-derived 50-day-old neurons with human fetal brain.
DOI: https://doi.org/10.7554/eLife.37382.007

## GIN progenitors generated via *LHX6* OE produce SST and PV neurons in vivo

To further investigate whether the population of human SST and PV interneurons could be enriched in grafts, we transplanted 7-week hPSC-derived MGE progenitors into the ventral forebrain of neonatal severe combined immunodeficiency (SCID) mice (*Figure 5a*). At 3 months post-transplantation, most grafted cells had become post-mitotic, and the NESTIN+ and KI67+ cells were rarely found (*Figure 5—figure supplement 1a*). Nearly 80% of hN+ cells were TUJ-1+ (*Figure 5—figure supplement 1a*), and most of them (74.92 ± 3.17% in the *LHX6* OE group and 72.21 ± 1.91% in the control group) were GABA+ (*Figure 5b*). Notably, the percentage of LHX6+ cells among the grafted *LHX6* OE cells, which maintained their identity in vivo, was higher than that among control cells at 3 months post-transplantation (*Figure 5c*). At 3 months post-transplantation, we found that the percentage of CR+ neurons was similar in the mice transplanted with *LHX6* OE or in control cells. However, the percentage of SST+ neurons was higher while the percentage of CB+ neurons was lower in the mice transplanted with the *LHX6* OE GINs than in controls (*Figure 5e and f*). Although the percentage of PV+ neurons in both groups was lower than that in other cell types in vivo, the number of human PV+ neurons in the *LHX6* OE group (60 PV + cells among 4524 hN+ cells) was significantly higher than that in controls (31 PV+ cells among 6608 hN+ cells). Thus, the GINs that were generated by *LHX6* OE promoted the generation of SST and PV neurons after transplantation into mice brain.

## Grafted neurons display characteristic firing patterns in vivo

Cortical SST neurons have several distinct kinds of firing properties, including regular-spiking, low threshold *spike* (LTS) and brusting firing patterns (*Yavorska and Wehr, 2016*). The characteristic feature of the cortical PV neurons is their fast-spiking action potentials (*Hu et al., 2014*). We assessed the membrane properties of our in-vitro-generated GIN subtypes using whole-cell patch-clamp recordings on enhanced green fluorescent protein (EGFP)-expressing neurons at 3 months and 5 months after transplantation (*Figure 6a*). We found a more negative resting membrane potential (RMP) and reduced input resistance in grafted neurons at 5 months compared to 3 months (*Figure 6b,c*). Action potential (AP) thresholds were not significantly different between the two time points (*Figure 6d*), but AP amplitude and fast after-hyperpolarization (fAHP) were increased in the 5-month grafted neurons in comparison with 3-month grafted neurons (*Figure 6—figure supplement 1*). These results suggest that the grafted neurons became more mature in vivo as the post-transplantation survival time is prolonged.

At 5 months post-transplantation, 6 of the 12 recorded grafted human neurons showed regular-spiking firings as well as an adaptation in neuronal firings (*Figure 6e*), which are the characteristics of neuronal firing for SST interneurons (*Yavorska and Wehr, 2016*). Notably, the rest of the recorded neurons (6 in 12 neurons) displayed fast-spiking-like firings and non-adaptation in neuronal firings as well as deep fAHPs (*Figure 6f*), which is thought to be typical of PV interneurons (*Nassar et al., 2015*; *Povysheva et al., 2013*). Next, we examined whether grafted human neurons have been innervated by endogenous neurons in the mouse brain. Spontaneous postsynaptic currents (sPSCs) were recorded for the 5-month grafted human neurons at holding potentials of −70 mV and −40 mV. As we expected, grafted neurons received robust inputs from other neurons in vivo with a spontaneous postsynaptic current (sPSC) frequency of 0.40 ± 0.076 Hz and an amplitude of 14.2 ± 0.62 pA at a holding potential of −70 mV (*Figure 6g,i,j*). It is worth noting that most synaptic inputs on grafted neurons are GABAergic inhibitory because a large number of the sPSCs became upward at a holding potential of −40 mV maintained using a recording pipette solution

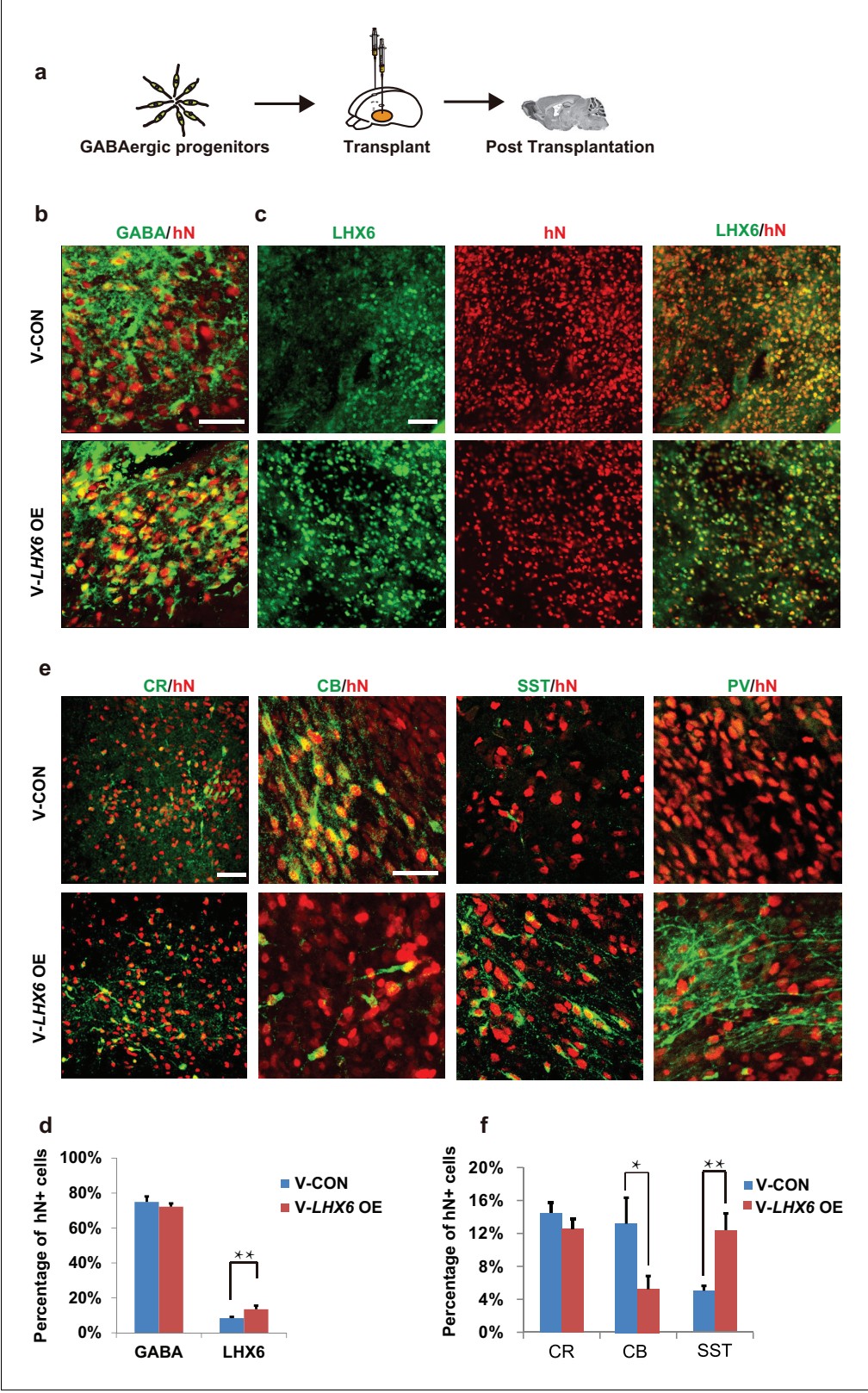

**Figure 5.** Neonatal transplantation showed that overexpression of *LHX6* increased the differentiation of SST and PV neurons. (a) Schematic showing the transplantation of hPSC-derived GABAergic progenitors into the basal forebrain of neonatal mice. (b–c) Immunostaining of GABA (b) and LHX6 (c) co-labeled with hN in grafted cells at 3 months after transplantation. Scale bar, 50 μm. (d) Quantification of GABA+ and LHX6+ cells among hN+ cells. Over 6000 hN+ cells were counted; n = 4 for V-CON and n = 5 for V-*LHX6* OE. (e) At 3 months after transplantation, all four GIN subtypes (CR, CB, SST,
*Figure 5 continued on next page*

*Figure 5 continued*

and PV) could be detected. Scale bar, 50 µm. (f) Quantification of GIN subtypes at 3 months after transplantation. Over 3000 hN+ cells were counted for each subtype; n = 4 for V-CON and n = 5 for V-*LHX6* OE.

DOI: https://doi.org/10.7554/eLife.37382.008

The following figure supplement is available for figure 5:

**Figure 5—1Figure supplement 1.** Electrophysiological characteristic of grafted human neurons.

DOI: https://doi.org/10.7554/eLife.37382.009

containing Cl⁻ with a reversal potential of −79 mV (*Figure 6h*). These results suggest that the grafted human neurons were fully functionally integrated into the existing endogenous network in the mouse brain 5 months after transplantation.

## Discussion

Here, we describe the application of a single transcription factor, *LHX6*, in combination with the SHH activator to robustly generate PV and SST interneuron subtypes from hESCs and iPSCs. A cell population in which PV and SST neurons together made up nearly 50% of the population could be yielded with stable *LHX6* overexpression and ventral patterning. Differentiation of human cortical interneurons in vitro follows the principle of early generation of CB and CR neurons and late generation of SST and PV neurons (*Cao et al., 1996*), which is consistent with the human fetal tissue research showing that SST and PV expression has been found at the beginning of middle-late gestation (*Zecevic et al., 2011*).

Effective generation of human GINs (75–90% efficiency) has been reported by several groups, including ours (*Liu et al., 2013a*; *Maroof et al., 2013*; *Nicholas et al., 2013*). However, the generation of PV and SST neurons was low in those studies and required co-culture with rodent cortical glial/excitatory neurons after fluorescence-activated cell sorting (FACS) (as listed in *Supplementary file 1*). The main reason may be the late and low-level expression of the LHX6 protein (*Maroof et al., 2013*; *Nicholas et al., 2013*). Our study provides a simple method to yield transplantable SST and PV interneurons in vitro, without the need for cell sorting or a co-culture system, indicating the that the expression of LHX6 is sufficient for the induction of SST and PV neurons.

Recently, LHX6 was reported to be one of the inducers for converting non-neuronal cells into GINs (*Yang et al., 2017*; *Colasante et al., 2015*; *Sun et al., 2016*) in vitro (Supplementary T1). Three factors (Ascl, Dlx2, and Lhx6) together with miR-9/9*−124 induced hESCs/iPSCs to become GINs (including the CB, CR, SST and Never Peptide Y (NPY) subtypes), but few PV neurons were observed at day 70 post-differentiation, without the treatment of ventral inducers (SHH or SAG) (*Sun et al., 2016*). In another study, the expression of ASCL1and DLX2 were induced in the hPSCs for 5 weeks to generate GINs (*Yang et al., 2017*). However, the expression of PV neurons was not detected. In our research, we found that *LHX6* alone was able to regulate the generation of SST and PV interneurons from hESCs/iPSCs with or without the treatment of the ventral inducers.

In summary, our current study indicated that LHX6 alone is essential and sufficient to enrich the population of PV and SST GIN subtypes, providing an efficient method for producing human PV/SST neurons in vitro. Our method offers the possibility to study the etiology of PV-/SST-relevant psychiatric diseases and drug discoveries.

## Materials and methods

### Human pluripotent stem cell culture and neural differentiation

hESCs (H9, Passage 40–60, WiCell Agreement NO.16-W0060, mycoplasma contamination testing see Supplementary T4) and iPSCs (ihtc, Passage 10–20, established in our laboratory) were maintained on vitronectin-coated plates (Life Technologies) with Essential eight medium, which was changed daily. Cells were passaged every 5 days through ethylenediaminetetraacetic acid (EDTA) (Lonza) digestion. For neural differentiation, hPSCs were detached by dispase (Life Technologies) to form embryoid bodies (EBs) and then cultured in neural induction medium (NIM) as previously described (*Yuan et al., 2015*). After floating for 7 days, EBs were attached on vitronectin-coated

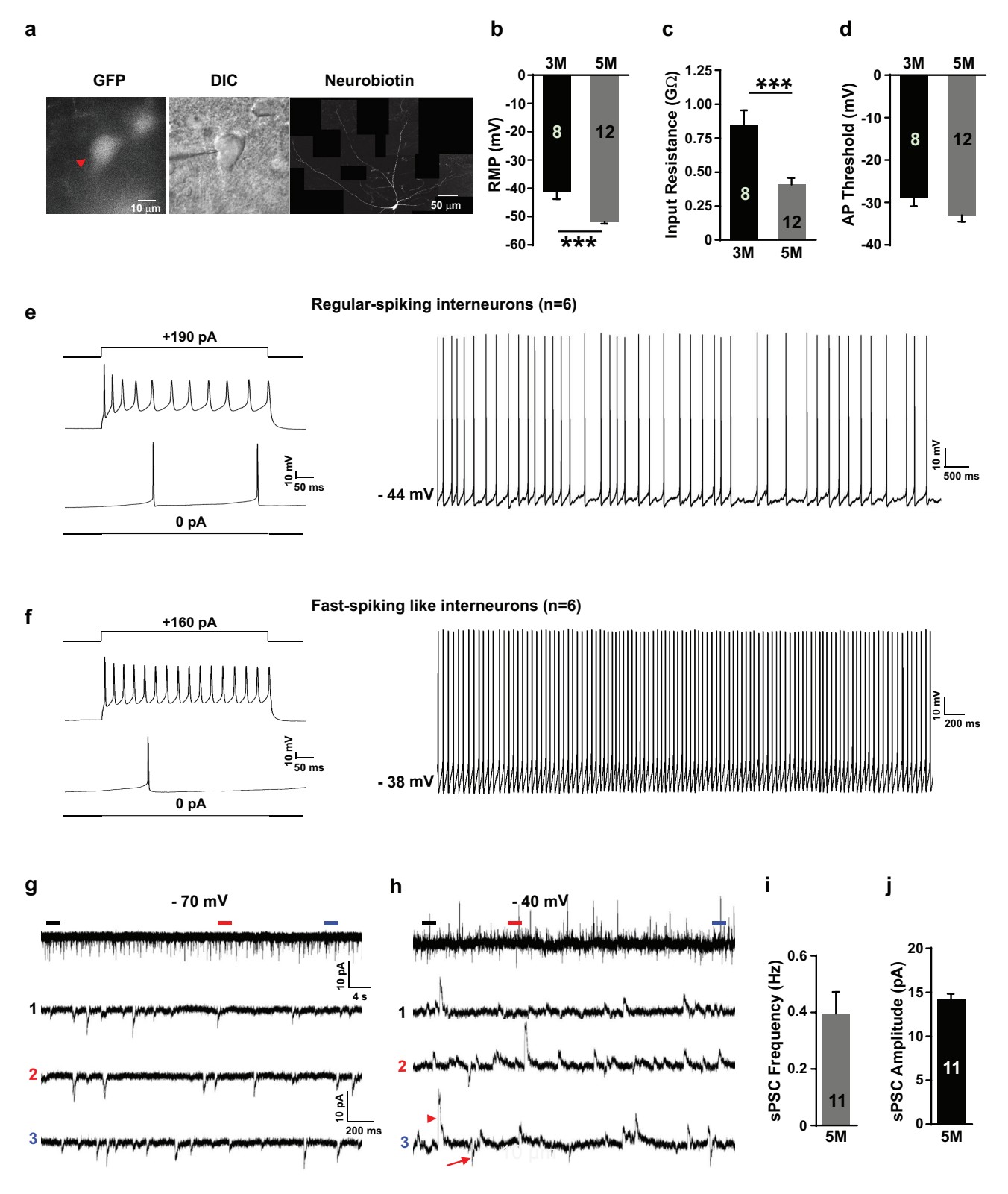

**Figure 6.** Grafted neurons show functional maturation and fast-spiking-like interneuron high-frequency firing in vivo. (a) Representative image of an *LHX6*-EGFP OE cell at 5 months after in vivo transplantation. The cell was recorded with whole-cell configuration and then filled with neurobiotin. The recorded cell was visualized with EGFP (left, red arrow) and differential interference contrast microscopy (DIC) (middle) and was reconstructed via neurobiotin staining (right). (**b–d**) Summary of resting membrane potential (RMP) (**b**), input resistance (**c**), and action potential (AP) threshold (**d**) from

*Figure 6 continued on next page*

*Figure 6 continued*

*LHX6*-EGFP OE cells at 3 months and 5 months after transplantation. ***, p<0.001. (**e**) Sample traces of voltage changes in an *LHX6*-EGFP OE cell. Changes of membrane potential were evoked by current injection in 0 pA and 190 pA, respectively (*left*). Spontaneous firings of an *LHX6-EGFP* OE cell at a subthreshold holding of −44 mV. (**f**) Sample traces of voltage changes in a *LHX6-EGFP* OE cell. Changes of membrane potential were evoked by current injection in 0 pA and 160 pA (*left*). The recorded cell could fire action potentials at a maximus rate of 40 Hz. Spontaneous firings of an *LHX6-EGFP* OE cell at a subthreshold holding of −38 mV (*right*). (**g–h**) Sample traces of spontaneous postsynaptic currents (sPSCs) at a holding potential of −70 mV (**g**) and −40 mV (**h**) from an *LHX6-EGFP* OE cell at 5 months after transplantation. The bottom three traces are enlarged from the top traces. (**i–j**) Summary of frequency (**i**) and amplitude (**j**) of sPSCs recorded at a holding potential of −70 mV from *LHX6-EGFP* OE cells at 5 months after transplantation.

DOI: https://doi.org/10.7554/eLife.37382.010

The following figure supplement is available for figure 6:

**Figure supplement 1.** Immunostaining of garfted cells.

DOI: https://doi.org/10.7554/eLife.37382.011

surfaces. Rosette structures could be observed at day 10–16. At day 16, rosette clones were detached manually with a 1 ml pipette. Non-neuroepithelial clones were removed at this stage. Neurospheres were continuously floated in NIM and then dissociated by TrypLE (Life Technologies) and plated on vitronectin (Life Technologies) and poly-l-ornithine (Sigma) pre-coated coverslips for further neuronal differentiation. For dorsal differentiation, no morphogen was added. For ventral differentiation, 500 nM SAG was added from day 10 to day 25. In all in vitro experiments examining the induction of LHX6 overexpression, dox was added at 3 µg/mL from day 10 to day 25 for continuous treatment.

## Human pluripotent stem cell electroporation

hPSCs were maintained under feeder-free conditions. hiPSCs or hESCs were treated with Rho Kinase (ROCK) inhibitor 2 hr before electroporation. Cells ($1 \times 10^6$) were dissociated into single cells by treating with EDTA for 5 min in an incubator and then were mixed with plasmids (OE: 20 µg donor plasmid and 5 µg TALEN arms) using the Lonza Nucleofector kit (VPH-5002). The cell mixture was electroporated in a Lonza Nucleofector 2b with the A023 program. After electroporation, the cells were quickly reseeded onto a vitronectin-coated six-well plate in Essential eight medium with ROCK inhibitor added for the first 24 hr. Stable colonies were selected after 5–7 days of continuous 0.5 µg/µl puromycin treatment.

## Immunostaining for cells and brain slices

Cells cultured on coverslips were fixed in cold fresh 4% paraformaldehyde for 30 min and rinsed three times with phosphate buffered saline. Cells were treated with 0.2% TritonX-100 for 10 min and blocked in 10% donkey serum for 1 hr (brain slices: 0.5% TritonX-100% and 5% donkey serum for 1 hr). Cells/brain slices were incubated at 4°C overnight in primary antibody diluted with 0.1% triton and 5% donkey serum. On the second day, cells/brain slices were incubated in secondary antibody diluted in 5% donkey serum for 30 min at room temperature. Coverslips were mounted for fluorescent imaging solution. The primary and secondary antibodies are listed in Supplementary T2.

## Quantitative real-time PCR

Total RNA was extracted in Trizol reagent (Invitrogen) as previously described (*Liu et al., 2013b*), and cDNA was reverse-transcribed by using the SuperScript III First-Strand kit (Invitrogen). RT-PCR was performed using the Bio-Rad MyiQ real-time PCR detection system. The primers used are listed in Supplementary T3.

## Animals and neonatal transplantation

SCID mice were purchased from the Model Animal Research Center of Nanjing University. All of the animal experiments followed standard experimental protocols and were approved by the Animal Care and Use Committee at Nanjing Medical University. The postnatal day (P0) SCID pups were randomly divided into two groups and were injected with 1 µl of day-35 V-CON/V-*LHX6* OE MGE-like precursors at a density of $1.5 \times 10^5$ cells/µl. hPSC-derived precursors were broken into little clusters using a Pasteur pipette technique, which yielded small spheres with diameters of around 30 µm 2–3

days before transplantation (*Liu et al., 2013b*). Cells were suspended in 10–20 µl NIM with B27 (Life Technologies) and penicillin (Life Technologies) on the day of transplantation. Both hemispheres of each neonatal pup were injected with 1 µl cell suspension using a glass micropipette. The injection site was located in the basal forebrain, in the middle between bregma and interaural line, about 1 mm lateral to the middline, at approximately 3 mm depth (*Xu et al., 2010*). After injection, the pups were laid on a pre-warmed cushion for 15 min and then were returned to their cage.

### RNA-sequencing analysis

Whole RNA-sequencing analysis was performed by the Beijing Genomics Institute. Control cells were ventralized with SAG. Day-50 neurons from the V-CON/V-*LHX6* OE groups were collected in Trizol. Total RNAs were extracted from the Trizol, and the single-end (50 bp) sequencing was performed on a HiSeq2000 platform (Illumina). RNA-seq reads were aligned to the human genome (GRCh37/hg19) using the software HISAT2 (DOI: 10.1038/nmeth.3317). Transcript abundance was quantified as FPKM (fragments per kilo base of exon per million fragments mapped). Differentially expressed genes were determined by DESeq2 (DOI: 10.1186/s13059-014-0550-8). Enriched Gene Ontology (GO) terms were identified with GSEA (DOI: 10.1073/pnas.0506580102).

### Slice preparation and electrophysiological recording

The brains of 3-months or 5-months post-transplantation mice were removed and sliced into 350 µm coronal sections in ice-cold NMDG (N-Methyl-D-glucamine)-containing solution consisting of (in mM): 93 NMDG, 93 HCl, 2.5 KCl, 1.2 $NaH_2PO_4$, 30 $NaHCO_3$, 20 HEPES, 25 glucose, 5 sodium ascorbate, 2 thiourea, 3 sodium pyruvate, 10 $MgSO_4$ and 0.5 $CaCl_2$, pH 7.35 bubbled with 95% $O_2$/5% $CO_2$ using a Vibratome 2000 (Leica Microsystems). Slices were incubated at 34°C for 10–15 min in oxygenated NMDG solution and then transferred to the artificial cerebrospinal fluid (ACSF), which contained (in mM): 126 NaCl, 4.9 KCl, 1.2 $KH_2PO_4$, 2.4 $MgSO_4$, 2.5 $CaCl_2$, 26 $NaHCO_3$, and 10 glucose, pH 7.4 at room temperature for about 0.5–1 hr before being transferred to a recording chamber containing ACSF bubbled with 95% $O_2$/5% $CO_2$ at 32°C. An upright fixed stage microscope (Olympus) equipped with epifluorescence and infrared-differential interference contrast (DIC) illumination, a charge-coupled device camera, and two water immersion lenses (10X and 60X) were used to visualize and target GFP-positive grafted cells. Glass recording electrodes (10–15 MΩ resistance) were filled with an intracellular solution consisting of (in mM): 136 K-gluconate, 6 KCl, 1 EGTA, 2.5 $Na_2ATP$, 10 HEPES (295 mOsm, pH = 7.25 with KOH). Whole-cell patch-clamp recordings were performed using an Olympus microscope (BX51WI) and data were collected and analyzed using the Axopatch1500B amplifier and pCLAMP10 software (Molecular Devices). Neurobiotin tracer (Vectorlabs, SP-1120) was delivered into cells for the identification of cell morphology through recording pipettes. Membrane potentials were maintained at around −70 mV, and a series of step currents with incremental amplitude (10 steps, with 10 pA or 5 pA increment) were injected to elicit action potentials. Input resistance was calculated from the slope of a current-voltage plot of the change in membrane voltage evoked by a series of current injection steps in 10 pA increments. Data analysis was performed using Clampfit and GraphPad. After recording, brain slices were postfixed with 4% paraformaldehyde overnight, and were then subjected to immunostaining for neurobiotin (Alexa Fluor 647 Streptavidin, S21374).

### Cell counting and statistical analysis

Images were acquired using an Eclipse 80i fluorescence microscope and a ZEISS LSM 700B confocal microscope. The quantification of fluorescent images was analyzed by Image J. For each cell line, more than 1500 cells for each index were counted, and at least three duplications were performed in each experiment. The number of nuclei labeled by Hoechst in each field was referred to as the total cell number for in vitro experiments, and the number of nuclei labeled by hN was referred to as the total grafted cell number in the transplantation experiment. More than 2000 hN+ cells were counted in each index (n (V-CON) = 4, n (V-*LHX6* OE)=5).

Data were compared by Student's *t*-test, one-way ANOVA, and two-way ANOVA. All graphical data were presented as mean ± SEM. Values were considered statistically significantly different at $p<0.05$(*), $p<0.01$(**), and $p<0.001$(***).

## Acknowledgments

We thank Dr. Zhang Zhengang (Institute of Brain Science at Fudan University) for kindly providing the LHX6 antibody and Dr. Su-Chun Zhang (Waisman Center and WiCell Research Institute at the University of Wisconsin) for kindly providing the AAVS1-TALEN plasmids. This study was supported by the Strategic Priority Research Program of the Chinese Academy of Sciences (Grant No. XDA16010306), the National Natural Science Foundation of China (Grants 81471301 and 31671063), the National Key Research and Development Program of China (2016YFC1306703), the Jiangsu Outstanding Young Investigator Program (BK20160044), the ShanghaiTech Start-up Foundation and the Jiangsu Province's Innovation Program, Postgraduate Research and Practice Innovation Program of Jiangsu Province.

## Additional information

### Funding

| Funder | Grant reference number | Author |
|---|---|---|
| National Natural Science Foundation of China | 81471301 | Yan Liu |
| National Key Research and Development Program of China | 2016YFC1306703 | Yan Liu |
| Jiangsu Outstanding Young Investigator Program | BK20160044 | Yan Liu |
| ShanghaiTech Start-up Foundation | | Shuijin He |
| Postgraduate Research and Practice Innovation Program of Jiangsu Province | | Fang Yuan |
| Chinese Academy of Sciences | Strategic Priority Research Program XDA16010306 | Yan Liu |
| National Natural Science Foundation of China | 31671063 | Yan Liu |

The funders had no role in study design, data collection and interpretation, or the decision to submit the work for publication.

### Author contributions

Fang Yuan, Conceptualization, Resources, Formal analysis, Methodology, Writing—original draft, Writing—review and editing; Xin Chen, Resources, Data curation, Software, Methodology; Kai-Heng Fang, Software, Formal analysis, Validation; Yuanyuan Wang, Software, Formal analysis; Mingyan Lin, Software, Writing—original draft; Shi-Bo Xu, Hai-Qin Huo, Formal analysis; Min Xu, Lixiang Ma, Yuejun Chen, Resources; Shuijin He, Resources, Writing—original draft, Writing—review and editing; Yan Liu, Conceptualization, Resources, Supervision, Funding acquisition, Validation, Writing—original draft, Writing—review and editing

### Author ORCIDs

Yan Liu http://orcid.org/0000-0003-2918-5398

### Ethics

Human subjects: H9 cell line was purchased from Wicell Agreeement (NO.16-W0060) For ips cell line, verbal and written consent was obtained from the person. The study was approved by the ethic community of Nanjing Medical University([2017]NO.217).

Animal experimentation: SCID mice were purchased from the Model Animal Research Center of Nanjing University. All of the animal experiments followed standard experimental protocols and were approved by the Animal Care and Use Committee at Nanjing Medical University (IACUC-1601129).

Decision letter and Author response
Decision letter https://doi.org/10.7554/eLife.37382.026
Author response https://doi.org/10.7554/eLife.37382.027

## Additional files

### Supplementary files

• Supplementary file 1. Supplementary Table 1: Summary of PV and SST neurons deriverd from hPSCs.
Supplementary Table 2: Antibodies used in this study.
Supplementary Table 3: Primers used in this study.
Supplementary Table 4: Mycoplasma contamination testing of cell lines.
DOI: https://doi.org/10.7554/eLife.37382.012

• Transparent reporting form
DOI: https://doi.org/10.7554/eLife.37382.013

### Data availability

Sequencing data have been deposited in GEO under accession codes GSE114553. All data generated or analysed during this study are included in the manuscript and supporting files.

The following dataset was generated:

| Author(s) | Year | Dataset title | Dataset URL | Database, license, and accessibility information |
|---|---|---|---|---|
| Fang Y, Fang K, Chen X, Wang Y, Lin M, Xu S, Huo H, Xu M, Ma L, Chen Y, He S, Liu Y | 2018 | Induction of human SST and PV neurons by expressing a single transcription factor LHX6 | https://www.ncbi.nlm.nih.gov/geo/query/acc.cgi?acc=GSE114553 | Publicly available at the NCBI Gene Expression Omnibus (accession no: GSE114553). |

The following previously published dataset was used:

| Author(s) | Year | Dataset title | Dataset URL | Database, license, and accessibility information |
|---|---|---|---|---|
| Blue B. Lake, Rizi Ai, Gwendolyn E. Kaeser, Neeraj S. Salathia, Yun C. Yung, RuiLiu, Andre Wildberg, Derek Gao, Ho-Lim Fung, Song Chen, Raakhee Vijayaraghavan, Julian Wong, Allison Chen, Xiaoyan Sheng, Fiona Kaper, Richard Shen, Mostafa Ronaghi, Jian-Bing Fan, Wei Wang, Jerold Chun, and Kun Zhang | 2016 | Single Cell Analysis Program - Transcriptome (SCAP-T) | https://www.ncbi.nlm.nih.gov/projects/gap/cgi-bin/study.cgi?study_id=phs000833.v3.p1 | Publicly available at the NCBI dbGaP (accession no. phs000833.v3.p1). |

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
