## [Decision Letter]

Thank you for submitting your article "Induction of human SST and PV neurons by expressing a single transcription factor *LHX6*" for consideration by *eLife*. Your article has been reviewed by two peer reviewers, one of whom is a member of our Board of Reviewing Editors, and the evaluation has been overseen by K VijayRaghavan as the Senior Editor. The reviewers have opted to remain anonymous.

The reviewers have discussed the reviews with one another and the Reviewing Editor has drafted this decision to help you prepare a revised submission.

Summary:

In this study, Yuan et al. describe the generation of somatostatin (SST) and parvalbumin (PV) GABAergic interneurons from human pluripotent stem cells (hPSC) using a directed differentiation approach based on overexpression of *LHX6*. These two types of interneurons play significant roles in neurological and psychiatric disorders, so the availability of a cell culture model system to study their function could represent an important advance in the field. It has been difficult to produce these types of interneurons using either reprogramming or directed differentiation of hPSC. The authors describe a relatively efficient system for production of the cells, and characterize the neurons through RNA-seq, immunofluorescence microscopy, transplantation in vivo and electrophysiological recording.

Essential revisions:

1) Subsection “*LHX6* promotes the generation of GINs” Figure 2B – what are the actual proportions of these cell types in control cultures?

2) Subsection “*LHX6* promotes the generation of GINs” Figure 2B – the cell types that were counted make up at most 40% of the total. Do the authors know what the remaining cells represent?

3) Figure 2D – what proportions of SST and PV positive cells develop the characteristic morphology as shown?

4) Figure 4 – were the controls for this study ventralised with SAG?

5) Subsection “Grafted neurons display characteristic firing patterns in vivo” – a reference to the electrophysiological properties of SST and PV neurons is missing.

6) "For neural differentiation, hPSCs were detached by dispase

(Life Technologies) to form embryoid bodies (EBs) and then cultured in neural induction medium (NIM) as previously described." What is the reference?

7) "dox was added at 3 μg/mL on day 10" – was this continued or just a one-day treatment? It is mentioned that in some experiments dox was continued from day 10 to day 25, but for some other experiments it is not clear.

8) "hPSC-derived progenitors were broken into little clusters with diameters of around 30 μm 2-3 days before transplantation". This statement needs clarification as to what is meant by "broken into" – was this done at a passage, what exactly was the procedure?

On a related note, are the cells mitotic, and if so for how, after forced *Lhx6* expression? While it has not been definitively shown for human, in mice *Lhx6*+ cells are post-mitotic and thus would not be referred to as "progenitors", but could be referred to as post-mitotic precursors.

9) "The injection site was located at the basal forebrain." This needs to be defined more precisely with coordinates.

10) The specificity of the *Lhx6* antibody needs to be established. Same for the *Foxg1* antibody, and the hN antibody.

11) Figure 3H. and also text referring to Figure 2. There is nothing distinctly chandelier like about dendritic morphology. The chandelier interneuron is defined by its distinctive axo-axonic targeting onto the initial segment of pyramidal neuron axons. Unless axo-axonic targeting is shown any mention of chandelier cell in results in a misrepresentation of the data.

12) Figure 5. What was the protocol used in this experiment regarding duration and timing of dox?

13) Where are the cells – in striatum? There are PV+ and *Lhx6*+ GABA projection neurons in the globus pallidus – could some of these transplanted cells have this fate?

14) The histology in Figure 5 looks odd – it seems that nearly all cells in the field express hN, but if these were "real" mge-like interneuron precursors they should migrate and disperse. Perhaps they need to try making transplants earlier in the protocol? Place them in the neocortex?

15) The trace in Figure 6 appears to be about 20Hz. This is not typically considered fast-spiking, which should be 40+ Hz in vitro.

16) It seems as though cells were cultured without pyramidal neurons. If that is the case any apparent PV expression is a bit hard to interpret since PV is at least partially activity dependent, depending on NMDA activation and ERBB4, for example, which one might not expect to see in monocultures of GABA cells. Please clarify.

---

## [Author Response]

Essential revisions:1) Subsection “LHX6 promotes the generation of GINs” Figure 2B – what are the actual proportions of these cell types in control cultures?

In control cultures, CB+, PV+ or SST+ cells were barely observed. CR+ neurons was yielded about 10% population. The proportion of interneuron subtypes are low without ventral patterning factors (SHH), which is understandable since default differentiation mostly generate glutamatergic projection neurons.

2) Subsection “LHX6 promotes the generation of GINs” Figure 2B – the cell types that were counted make up at most 40% of the total. Do the authors know what the remaining cells represent?

Since we do not add any ventral patterning factors, the remaining cells should be glutamatergic neurons and neural progenitor cells since there are still 20-40% cells express dorsal forebrain marker PAX6+ at beginning (Figure 1G in manuscript). Identity of neural progenitors was also confirmed by *SOX2* staining.

Besides, the cortical marker TBR2 (day 50) and glutamatergic neuron marker vGlut1 (day120), (Author response image 1). In this circumstance, the remaining cells were neural progenitors and glutamatergic neurons.

**Author response image 1. respfig1:** Cells in *LHX6* OE group expressing *SOX2*, TBR2 and vGlut1 (scale bar, 100μm) (related to Figure 2 in manuscript).

3) Figure 2D – what proportions of SST and PV positive cells develop the characteristic morphology as shown?

We set the morphology that the neuron having more than 5 secondary branches as characteristic morphology. About 74% (193 of 261 SST+ neurons) of SST neurons have the characteristic morphology at day 45. About 30% of PV neurons (32 of 106 PV+ neurons) have the characteristic morphology at day 80.

Description of “We set the morphology that the neuron having more than 5 secondary branches as characteristic morphology. Most SST neurons have the characteristic morphology since day 45. About 30% of PV neurons have the characteristic morphology at day 80.” was added in the manuscript.

4) Figure 4 – were the controls for this study ventralised with SAG?

We apologize for the misunderstanding. Yes, the controls were ventralised with SAG. We added the description “Control cells were ventralised with SAG.” in this revision.

5) Subsection “Grafted neurons display characteristic firing patterns in vivo” – a reference to the electrophysiological properties of SST and PV neurons is missing.

We apologize for missing the reference. Reference was added in this revision (Hu, Gan, and Jonas, 2014).

6) "For neural differentiation, hPSCs were detached by dispase(Life Technologies) to form embryoid bodies (EBs) and then cultured in neural induction medium (NIM) as previously described." What is the reference?

We apologize for missing the reference. Reference was added in this revision (Yuan et al., 2015).

7) "dox was added at 3 μg/mL on day 10" – was this continued or just a one-day treatment? It is mentioned that in some experiments dox was continued from day 10 to day 25, but for some other experiments it is not clear.

We apologize for the confusion. The treatment was continued to turn on *LHX6* overexpression from day 10 to day 25, which was consistent in all experiments. We added description “In all in vitro experiment of induction of *LHX6* overexpression, dox was added at 3 μg/mL from day 10 to day 25 for continuous treatment.” in the revision.

8) "hPSC-derived progenitors were broken into little clusters with diameters of around 30 μm 2-3 days before transplantation". This statement needs clarification as to what is meant by "broken into" – was this done at a passage, what exactly was the procedure?

The statement “broken into” means that we use Pasteur pipette technique for triturating neurospheres into small clusters instead of passage. We flame the end of a cotton-filtered Pasteur pipette and make the opening to 0.2-0.5mm in diameter. And then we used Pasteur pipette to break the sphere into small clusters with diameters of around 30 μm instead of passage (Liu et al., 2013).

On a related note, are the cells mitotic, and if so for how, after forced Lhx6 expression? While it has not been definitively shown for human, in mice Lhx6+ cells are post-mitotic and thus would not be referred to as "progenitors", but could be referred to as post-mitotic precursors.

Yes. After we forced the expression of *LHX6*, part of the cells still are mitotic. We found there were a proportion of cells co-express *SOX2/ LHX6* at day 35, and EdU/*LHX6* on day 50 inside the clusters (EdU treatment with 12 hours) (Author response image 2). It suggested that the cells might be a mixture population of progenitors and post-mitotic precursors. And thanks for the reminder, we changed *LHX6* related description "progenitors" to "precursors" in this revision.

**Author response image 2. respfig2:** *SOX2, LHX6*, EdU staining of day35 V-*LHX6* OE cells (scale bar, 100μm).

9) "The injection site was located at the basal forebrain." This needs to be defined more precisely with coordinates.

Thanks for the comment. The description has changed to“The injection site was located at the basal forebrain, in the middle between bregma and interaural line, about 1 mm lateral to the midline, approximately 3mm depth”, which was adapted from Xu et al., 2010.

10) The specificity of the Lhx6 antibody needs to be established. Same for the Foxg1 antibody, and the hN antibody.

*Lhx6* antibody is a kind gift from Dr. Yang Zhengang (Institute of Brain Science at Fudan University). And the antibody specificity was established by E18.5 mouse forebrain immunostaining (Author response image 3). Besides, we also adapted a figure of *LHX6* staining for E13.5 mouse forebrain from Dr. Yang Zhengang’s paper to establish the specificity of *Lhx6* antibody (Author response image 3) (Z. Liu et al., 2018, Cereb Cortex).

**Author response image 3. respfig3:** *LHX6* staining. (**a–b**) *LHX6* staining of E18.5 mouse forebrain cortex (scale bar, 100μm).

FOXG1 antibody specificity was confirmed by E18.5 mouse forebrain (Author response image 3). Besides, in our previous study, the same FOXG1 antibody was used to clarify the characteristic of forebrain (D6-12, FOXG1 positive) and midbrain (D1-12, FOXG1 negative) cells that derived from hESCs (Xi et al., 2012, Stem Cells. 2012 Aug; 30(8): 1655–1663). Also, the antibody was widely used in other studies including ours (Kim et al. 2014; Renner, Lancaster et al., 2017, The EMBO Journal;Liu et al., 2013).

**Author response image 4. respfig4:** FOXG1 staining. (**a–b**) FOXG1 staining of E18.5 mouse forebrain (scale bar,100μm).

Human nuclei antibody specificity was confirmed by human grafted cells in mouse brain slice. Only the grafted human cells expressed human nuclei, but the mouse endogenous cells are negative for human nuclei. Besides, this antibody is used by many transplantation studies (Guan et al., 2013, Biomaterials, 34(24), 5937-5946; Ma et al., 2012, Cell Stem Cell, 10(4), 455-464; Yang et al., 2010, PLOS ONE, 5(10), e13547; Liu et al., 2013,).

**Author response image 5. respfig5:** hN staining of transplanted mouse brain slice (scale bar, 100μm).

11) Figure 3H. and also text referring to Figure 2. There is nothing distinctly chandelier like about dendritic morphology. The chandelier interneuron is defined by its distinctive axo-axonic targeting onto the initial segment of pyramidal neuron axons. Unless axo-axonic targeting is shown any mention of chandelier cell in results in a misrepresentation of the data.

We apologize for the misrepresentation. We have changed the description “chandelier” to “multipolar” in the revision.

12) Figure 5. What was the procotol used in this experiment regarding duration and timing of dox?

The cells used in transplantation followed the ventral differentiation protocol in vitro (as described in Figure 3). *LHX6* OE group cells was treated with 3ug/ml from day 10 to day 25.

13) Where are the cells – in striatum?

Based on our data, most of the transplanted cells tend to disperse from the transplantation site after 3 months transplantation, as shown in Author response image 6. Some of the cells migrated out of the transplantation site (Author response image 6). There were scattered grafted cell in striatum as well as in cortex.

**Author response image 6. respfig6:** 3 months mouse brain slices (scale bar, 100μm).

There are PV+ and Lhx6+ GABA projection neurons in the globus pallidus – could some of these transplanted cells have this fate?

We observed sparse grafted cells in globus pallidus (Author response image 6). However, we did not see PV+ and *Lhx6*+ colabeled cells.

14) The histology in Figure 5 looks odd – it seems that nearly all cells in the field express hN, but if these were "real" mge-like interneuron precursors they should migrate and disperse. Perhaps they need to try making transplants earlier in the protocol? Place them in the neocortex?

We apologize for making the confusion. Most images we presented in Figure 5 are in the transplantation site that making it seems like nearly all cells in the field express hN. We add images that co-staining HO and HN at transplantation site to make it clear that not all the cells are positive for HN (Author response image 6). Furthermore, related to question 13, a lot of grafted cells migrated out of the graft site (Author response image 6).

15) The trace in Figure 6 appears to be about 20Hz. This is not typically considered fast-spiking, which should be 40+ Hz in vitro.

Thanks for the comments. Interneurons that fire action potential above 40 Hz are typically defined as fastspiking interneurons, so that we have named them as “fast-spiking-like firing” in the context. Besides, PV neurons might take longer time to be functional mature than other GABAergic interneurons subtype since PV neurons expressed at late term gestation in human fetal tissue (Zecevic, et al. 2011).

16) It seems as though cells were cultured without pyramidal neurons. If that is the case any apparent PV expression is a bit hard to interpret since PV is at least partially activity dependent, depending on NMDA activation and ERBB4, for example, which one might not expect to see in monocultures of GABA cells. Please clarify.

Thanks for the comment.

For PV expression, it might be directly driven by *LHX6*-overexpression even without pyramidal neurons. *Lhx6* might function to directly drive the specification of PV and SST neurons (5).

For activity part, we agree that “PV is at least partially activity dependent”. In our system, activity could at least come from three ways: (1) since GABA is excitable in the newborn neurons (Ben-Ari, 2002, Nat Rev Neurosci3(9): 728-739; Rivera, Voipio et al. 1999, Nature, 397: 251), GABAergic synapses that form among interneurons are the driver for neural activity; (2) our data showed that newly generated interneurons displayed a relative more positive resting membrane potential (around 40 mV, Figure 6B), where neurons could get excited more easily; (3) vGlut1 staining of day120 cells in V-LHX6 OE group revealed that there was a small amount of excitatory inputs in our system which could trigger the PV activity (less than 1%).

**Author response image 7. respfig7:** V-*LHX6* OE neurons also received a small amount of excitatory inputs by vGlut1.